# Dual-Phase Continual Learning: Supervised Adaptation Meets Unsupervised Retention

**Vaibhav Singh**  *vaibhav.singh@mila.quebec*
*Department of Computer Science*
*Mila, Concordia University*

**Rahaf Aljundi**  *rahaf.al.jundi@toyota-europe.com*
*Toyota Motor Europe*

**Eugene Belilovsky**  *eugene.belilovsky@concordia.ca*
*Mila, Concordia University*

**Reviewed on OpenReview:** *https://openreview.net/pdf?id=GFrHdXzZwo*

## Abstract

Foundational Vision-Language Models (VLMs) excel across diverse tasks, but adapting them to new domains without forgetting prior knowledge remains a critical challenge. Continual Learning (CL) addresses this challenge by enabling models to learn sequentially from new data while mitigating the forgetting of prior information, typically under supervised settings involving label shift. Nonetheless, abrupt distribution shifts can still cause substantial forgetting, potentially nullifying the benefits of supervised updates, especially when storing or replaying past data is infeasible. In this work, we propose leveraging unlabeled test-time data in an unsupervised manner to ***reinforce prior task performance without requiring replay or stored examples***. Unlike traditional Test-Time Adaptation (TTA), which primarily focuses on domain shift or corruption, our method improves performance on earlier tasks by exploiting representative test samples encountered during deployment. We introduce a simple Teacher-Student framework with gradient-based sparse parameter updates, and show that it effectively mitigates forgetting in class-incremental CL for VLMs, offering a memory-free alternative to episodic replay with strong empirical results.

## 1 Introduction

Foundation models in computer vision have shown impressive performance on various downstream tasks and domains, which renders them a key building block of various solutions, including generative vision language models (Li et al., 2022; Chen et al., 2023; Bommasani et al., 2021). However, naively adapting these pre-trained models to distribution shifts or new tasks often leads to *catastrophic forgetting* (McCloskey & Cohen, 1989), where new learning sessions interfere with what a model has previously acquired.

To address catastrophic forgetting, Continual Learning (CL) enables models to adapt to evolving data distributions over time. Key approaches include regularization-based methods (Kirkpatrick et al., 2017; Maltoni & Lomonaco, 2019; Li & Hoiem, 2017; Schwarz et al., 2018; Singh et al., 2024), external memory approaches (Lopez-Paz & Ranzato, 2017; Rolnick et al., 2019; Shin et al., 2017), and dynamic model architecture techniques (Douillard et al., 2022; Pham et al., 2021). However, training from scratch often overlooks the rich representations learned by large pre-trained models. With the advent of foundation models, there is a growing interest in integrating CL with their representational power (Han et al., 2021; Radford et al., 2021; Ridnik et al., 2021; Caron et al., 2021; Oquab et al., 2023).

Significant efforts have been made to improve foundational models' adaptability to new data streams (Wang et al., 2022d; Smith et al., 2023; Zhou et al., 2023; Zhang et al., 2023; Goyal et al., 2023; Wang et al.,

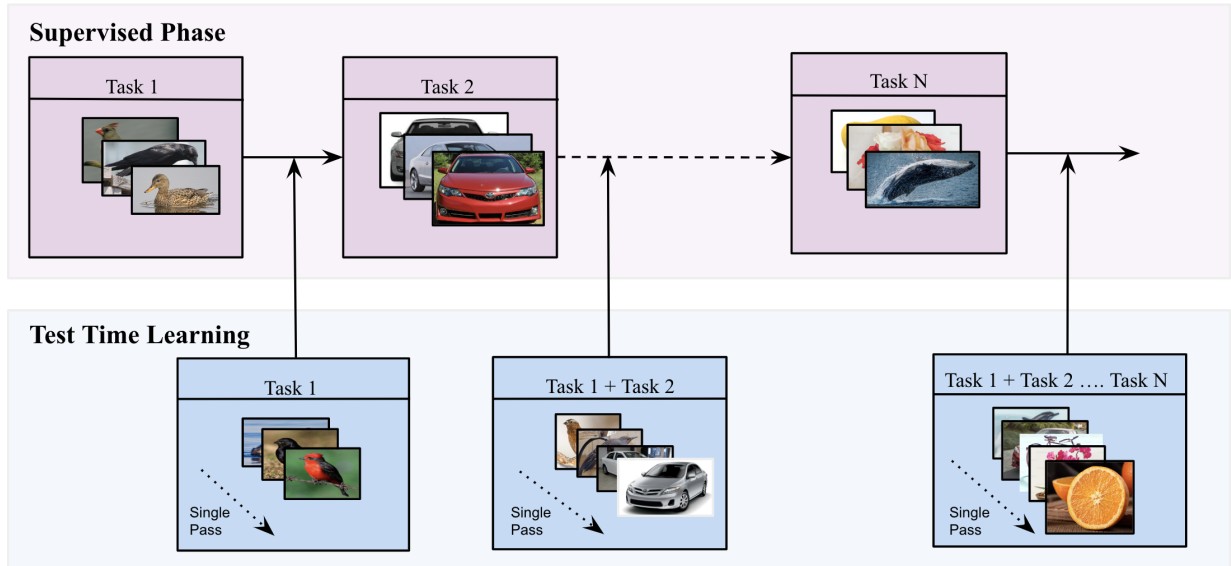

Figure 1: An illustration of our proposed setting of **Continual Learning with Interleaved Test Time Learning.** After each supervised training session, the model is deployed to adapt in an unsupervised deployment phase, where it encounters data from both current and previously seen tasks. During this phase, the model adapts to the current task's classes while striving to preserve performance on earlier tasks, thereby mitigating forgetting.

2022c), primarily through supervised training. However, this focus often leaves models static and prone to catastrophic forgetting (Wang et al., 2024a; Prabhu et al., 2023). In contrast, the diverse unsupervised data encountered during inference presents an underexplored opportunity to mitigate forgetting without additional supervision. We consider a continual learning setup, especially the challenging scenario of class incremental learning (CIL), where supervised training phases are interleaved with unsupervised deployment, as shown in Figure 1. To mitigate catastrophic forgetting, the model leverages unlabeled test-time data in an online manner—processing each sample once and discarding it, thereby reducing both privacy risks (Verwimp et al., 2023) and computational overhead.

To the best of our knowledge, we are among the early works that utilize test-time data for alleviating forgetting in continual learning, particularly through an interleaved test-time learning stage. We propose a novel Dual-Phase Continual Learning framework: **DoSAPP** (described in Section 3) that uniquely combines supervised adaptation with unsupervised retention, all without relying on replay buffers or explicit task boundaries. Built upon the CLIP foundation model (Radford et al., 2021), which offers strong generalization and transfer (Rasheed et al., 2023; Pei et al., 2023), our approach introduces two distinct learning phases: supervised sessions enable efficient task-specific adaptation through sparse parameter updates while unsupervised sessions promote long-term stability by reinforcing previously acquired knowledge. Central to this design is the Teacher-Student (Tarvainen & Valpola, 2017b) framework governed by our novel *dual-momentum* mechanism, which decouples the adaptation rates of the teacher and student to balance plasticity and stability. Additionally, cumulative mask consolidation across tasks ensures scalable memory retention without interference. This synergy enables robust continual learning in realistic, dynamically evolving environments.

Existing works such as Test-Time Adaptation (TTA) (Sun et al., 2020b; Wang et al., 2020; Zhang et al., 2022; Niu et al., 2022; Sun et al., 2020a) and Continual Test-Time Adaptation (CTTA) (Wang et al., 2022a; Gong et al., 2022; Niu et al., 2022; Song et al., 2023; Tian & Lyu, 2024; Wang et al., 2024b) similarly utilize unsupervised test-time data, but with a different focus: *adapting models to unknown distribution shifts in data during deployment.* These methods consider previously unseen domain shifts and corruptions in the test-time data itself, aiming to perform well on this unsupervised data, while not degrading the performance of the model on past data. Further, these methods cannot address forgetting in a class incremental learning (CIL) setup. On the other hand, our setting, which interleaves supervised and unsupervised sessions, considers leveraging test-time data that does not require domain shift/corruption with respect to previously seen data. Instead, it aims to use the unsupervised data to combat forgetting in the CIL setting through the supervised

Table 1: ***Comparison of Continual Test-Time Adaptation (CTTA) methods with our approach, DoSAPP***. Our work is fundamentally orthogonal to TTA and CTTA methods, which primarily tackle distribution shifts during test time. Moreover, existing CTTA approaches do not typically address forgetting of previous tasks (classes) upon the introduction of new ones. These methods are also incompatible with class incremental learning (CIL). In contrast, DoSAPP leverages test-time data to enhance CIL performance and reduce forgetting through unsupervised learning on test samples.

| Method | Corrupted Data | Uncorrupted Data | Interleaved Supervision | Reduces Forgetting | Handles New Classes (CIL) |
|---|---|---|---|---|---|
| CoTTA (2022) (Wang et al., 2022a) | ✓ | ✗ | ✗ | ✗ | ✗ |
| NOTE (2022) (Gong et al., 2022) | ✓ | ✗ | ✗ | ✗ | ✗ |
| EATA (2022) (Niu et al., 2022) | ✓ | ✗ | ✗ | ✓ | ✗ |
| EcoTTA (2023) (Song et al., 2023) | ✓ | ✗ | ✗ | ✓ | ✗ |
| RMT (2023) (Döbler et al., 2023) | ✓ | ✗ | ✗ | ✗ | ✗ |
| PSMT (2024) (Tian & Lyu, 2024) | ✓ | ✗ | ✗ | ✗ | ✗ |
| DSS (2024) (Wang et al., 2024b) | ✓ | ✗ | ✗ | ✗ | ✗ |
| **DoSAPP (ours)** | ✓ | ✓ | ✓ | ✓ | ✓ |

sessions. Table 1 highlights the key differences between our proposed approach **DoSAPP** and existing CTTA methods.

Our contributions are as follows: 1) We propose a new setting for continual learning where test-time data can be leveraged for forgetting without explicit knowledge of task boundaries, especially in the challenging scenario of CIL. 2) We investigate different baselines for this setting. 3) We propose a novel approach that illustrates the utility of test-time data in supervised continual learning and the significant reduction in forgetting without relying on any external replay buffer.

## 2  Related Work

**Continual Learning from Pre-trained Models:** Continual learning with pre-trained models is gaining popularity due to the availability of powerful foundation models (Radford et al., 2021; Oquab et al., 2023; Brown et al., 2020). Recent approaches (Koh et al., 2022; Boschini et al., 2022) employ a Teacher-Student framework with knowledge distillation but rely on memory buffers to mitigate catastrophic forgetting, which can be memory-intensive (Zhou et al., 2022; Prabhu et al., 2023). Additionally, stored logits become outdated, requiring updates with task boundary information, which may not always be available in task-free CL. Further recent works such as (Chen et al., 2025) utilize OOD-based techniques to calibrate only the classifier layer of a pretrained model using test-time samples, aiming to improve performance in CIL setting. While our method also operates on the test stream, it differs fundamentally from (Chen et al., 2025): their approach requires access to task boundaries, which may not be available at inference. In contrast, our method makes no such assumptions and remains applicable even when task boundaries are unknown.

To address these limitations, CLIP (Radford et al., 2021) presents an attractive alternative as it inherently avoids classification head issues and retains a broad feature space, making it well-suited for continual learning. Inspired by SPU (Zhang et al., 2024), which preserves generic knowledge by modifying a sparse subset of parameters based on gradient scoring, we explore leveraging test data in continual learning. This opens avenues for self-supervised techniques to enhance feature representations while mitigating recency bias.

**Test Time Adaptation (TTA):** TTA methods primarily focus on handling domain shifts and adapting to data corruption in test data. These approaches aim to improve model performance on the adapted test data, often leveraging techniques such as self-supervised learning (Sun et al., 2020a), batch normalization (Nado et al., 2020; Vianna et al., 2024), entropy minimization (Wang et al., 2020; Niu et al., 2023), pseudo labeling (Chen et al., 2022; Liang et al., 2021), and continually adapting to varying test-time distribution shifts (Wang et al., 2022a; Gong et al., 2022; Niu et al., 2022; Song et al., 2023; Tian & Lyu, 2024; Wang et al., 2024b). In contrast, our approach leverages test-time data for a broader purpose: mitigating forgetting of past tasks. Unlike TTA methods, which leverage the test-time data to correct domain shifts, our method utilizes unsupervised test-time data from previous tasks to ***retain past knowledge obtained in the supervised training sessions.***

## Supervised Continual Learning | Unsupervised Test Time Learning

Figure 2: DoSAPP employs Teacher-Student ($\mathcal{M}_T$, $\mathcal{M}_S$) models respectively. In the Supervised Continual Learning phase, $\mathcal{M}_S$ performs sparse parameter selection using a gradient-based scoring function $\mathcal{F}$, followed by training on the selected parameters $\boldsymbol{\theta}^{\mathbf{m}} \in \boldsymbol{\theta}^S$. After each update, $\mathcal{M}_T$ parameters $\boldsymbol{\theta}^T$ are updated through weighted exponential smoothing based on the affine projection of the boolean mask $\mathbf{m}$, controlled by dual momentum terms $\delta$ and $\gamma$ for $\mathcal{M}_T$ and $\mathcal{M}_S$, respectively. In the unsupervised test-time learning phase, $\mathcal{M}_S$ adapts using "pseudo-label" derived from $\mathcal{M}_T$-$\mathcal{M}_S$ logits comparison. $\mathcal{M}_T$ then undergoes weighted smoothing again, with momentum terms $\delta$ and $\lambda$ for $\mathcal{M}_T$ and $\mathcal{M}_S$ (where $\gamma < \lambda < \delta$). This two-phase approach ensures generalization over previous knowledge while maintaining adaptability to new tasks.

## 3 Methodology

We propose a novel class incremental continual learning (CIL) setting using test-time data. As shown in Figure 1, the model recovers lost knowledge from previously seen tasks after each supervised session, adapting to new classes while minimizing forgetting without relying on an external replay buffer. We consider a setting where supervised datasets $[\mathcal{D}_1^s, \mathcal{D}_2^s, .....\mathcal{D}_T^s]$ drawn from different distributions arrive incrementally at training sessions $t$ ranging from 0 to $T$. Each session $t$ with $N_t$ instances includes dataset $\mathcal{D}_t^s = (\mathbf{x}_i^t, y_i^t)_{i=1}^{N_t}$ where an instance $\mathbf{x}_i^t \in \mathbb{R}^D$ belongs to class $y_i^t \in Y_t$, with disjoint label spaces $Y_t \cap Y_{t'} = \phi$ for $t \neq t'$. At session $t$, only the current dataset $\mathcal{D}_t^s$ is available for training the model $\mathcal{M}(\boldsymbol{\theta})$.

After training on $\mathcal{D}_t^s$, the model is deployed until $\mathcal{D}_{t+1}^s$ becomes available. During this phase, it encounters unsupervised test-time data $\mathcal{D}_t^u$, drawn from all previously seen tasks as shown in Figure 1. We leverage this data for online unsupervised adaptation to mitigate forgetting. Evaluation is performed on distinct test datasets $\mathcal{D}_t^e$ to ensure proper generalization assessment.

We further note that although supervised phases may permit multiple passes through the data until convergence, it would be impractical to collect unsupervised data in production and then perform adaptation on it. We thus restrict the unsupervised phase to be in the online setting (Sun et al., 2020a; Jang et al., 2022; Cai et al., 2021). This is especially important in cases where data privacy is a constraint, e.g., an assistant robot in a private smart home environment.

### 3.1 DoSAPP: Double Smoothing via Affine Projected Parameters

We propose a simple yet effective method for continual test-time learning, Double Smoothing via Affine Projected Parameters, aka DoSAPP. Our approach combines two key components: **1) sparse and local updates:** to reduce forgetting, maintain generalization, by constraining adaptation to a small set of parameters, and **2) Teacher-Student framework** to promote stability in online updates and minimize forgetting. In the continual test time learning, we can identify two distinct phases of learning, as outlined below.

---

**Algorithm 1** DoSAPP algorithm for continual and test time learning

---

**Require:** $\mathcal{M}_S(\boldsymbol{\theta}^S)$, CLIP loss: $\mathcal{L}(.,.,.)$, sparsity threshold $c$, learning rate $\eta$.

1: $\boldsymbol{\theta}^T \leftarrow \boldsymbol{\theta}^S$
2: **for** $t$ in tasks **do**
3:     $\boldsymbol{\theta}^{\mathbf{m}} \leftarrow$ top-K (K=$c$) params from MLP layers of $\boldsymbol{\theta}^S$ based on $\mathcal{F}$           ▷ Sparse Selection, Eq. 1
4:     **for** $(x_i, y_i)$ in $\mathcal{D}_t^s$ **do**
5:        $\boldsymbol{\theta}^{\mathbf{m}} \leftarrow \boldsymbol{\theta}^{\mathbf{m}} - \eta\nabla\mathcal{L}(\mathcal{M}_S(x_i), y_i)$                          ▷ Take one SGD step
6:        $\boldsymbol{\theta}_{i+1}^T \leftarrow p\boldsymbol{\theta}_i^T + q\boldsymbol{\theta}_{i+1}^S$                ▷ Dual momentum for teacher EMA update, Eq 3
7:     **end for**
8:     Compute union of masks for all tasks seen so far $\mathbf{m}_u$           ▷ Start of Unsupervised Phase
9:     Select $\mathbf{m}_u$ params in $\mathcal{M}_S$
10:     **for** $x_i$ in $\mathcal{D}_t^u$ **do**
11:        $l_T \leftarrow \max(\mathcal{M}_T(x_i), \dim = 1)$
12:        $l_S \leftarrow \max(\mathcal{M}_S(x_i), \dim = 1)$
13:        **if** $l_T > l_S$ **then**
14:           $\hat{y} \leftarrow \arg\max(\mathcal{M}_T(x_i))$
15:        **else**
16:           $\hat{y} \leftarrow \arg\max(\mathcal{M}_S(x_i))$
17:        **end if**
18:        $\boldsymbol{\theta}^{\mathbf{m}_u} \leftarrow \boldsymbol{\theta}^{\mathbf{m}_u} - \eta\nabla\mathcal{L}(\mathcal{M}_S(x_i), \hat{y})$                      ▷ Take one SGD step
19:        $\boldsymbol{\theta}_{i+1}^T \leftarrow p'\boldsymbol{\theta}_i^T + q'\boldsymbol{\theta}_{i+1}^S$              ▷ Dual momentum for teacher EMA update, Eq 6
20:     **end for**
21: **end for**

---

### Phase 1: Continual Learning Supervised Training with Sparse Selected Parameters

Our goal is to rapidly acquire new knowledge while preserving generic knowledge during both training and testing. To achieve this, we update only a small subset (sparsity threshold $c = 10\%$ of parameters. We choose $c = 10\%$ for all main experiments because it offers a consistent efficiency–performance trade-off across tasks and ensures stable behavior during long horizon test time adaptation. It has been demonstrated that updating only a small subset of relevant parameters in pre-trained models like CLIP can significantly reduce forgetting (Zhang et al., 2024). Complementary findings indicate that the MLP blocks in transformers act as key-value memory components, with the first MLP layer serving as a pattern detector (Geva et al., 2020). This implies that updating only the first MLP layer may suffice for retaining prior knowledge. Thus, we restrict updates to the first MLP layer of each transformer block in CLIP. From these candidates, we select the top-K (K=$c$) parameters, where $c$ is the sparsity threshold. This results in sparse, localized parameter updates, instead of broad model changes which helps preserve prior knowledge. For the sake of simplicity, model parameters in the paper refer to these candidate parameters.

Following (Zhang et al., 2024), we use the gradient magnitude of the loss with respect to the input data to assess parameter relevance, where a higher magnitude indicates a greater expected reduction in loss. We optimize the model $\mathcal{M}_S$ by first identifying the most relevant parameters from the candidate parameters of first MLP layer of each transformer block, $\boldsymbol{\theta}^{\mathbf{m}}$ ($\boldsymbol{\theta}^{\mathbf{m}} \in \boldsymbol{\theta}^S$), using:

$$\mathcal{F}\left(\boldsymbol{\theta}_{ij}^S, \mathcal{D}_t^s\right) = \left\|\frac{1}{N_t'}\sum_{k=1}^{N_t'} g_{ij}\left(x_k\right)\right\|, \tag{1}$$

where $g_{ij}\left(x_k\right)$ is the gradient of the CLIP loss $\mathcal{L}(\mathcal{M}_S, x_k, y_k)$ w.r.t. parameter $\boldsymbol{\theta}_{ij}^S$, computed per data point $(x_k, y_k) \in \mathcal{D}_t^s$. Iterating over the dataset once, we compute these scores and select the top-K $(K = c)$ most relevant parameters based on a sparsity threshold $c = 0.1$. This results in a binary mask $\mathbf{m}$, freezing all but the selected parameters.

**Teacher Student Framework**

To ensure stability later during online updates and reduce forgetting, we utilize a Teacher-Student framework (Tarvainen & Valpola, 2017a; Koh et al., 2022; Boschini et al., 2022; Döbler et al., 2023; Michel et al., 2024) where the student model is denoted by $\mathcal{M}_S(\boldsymbol{\theta}^S)$ and the teacher model is denoted by $\mathcal{M}_T(\boldsymbol{\theta}^T)$.

During both supervised and unsupervised phases, the teacher model $\mathcal{M}_T$ parameters $\boldsymbol{\theta}^T$ are updated using the exponentially moving average (EMA) of the student model parameters $\boldsymbol{\theta}^S$. Typically, in the Teacher-Student framework, all teacher parameters move toward the student parameters with a single smoothing parameter (momentum). However, we demonstrate that a single smoothing parameter is insufficient and leads to suboptimal performance, as shown in our ablation studies (Table 6) and in the Appendix (Table 7). In our setting, most of the student model parameters (candidate parameters of the first MLP block) remain frozen, with only a small subset (sparsity threshold, $c = 10\%$) being updated. We propose that the teacher model parameters corresponding to frozen candidate parameters of the student model should adapt at a different rate than those associated with active parameters. To address this, we introduce dual smoothing parameters (dual momentum), adjusting teacher parameter updates based on an affine transformation of the binary mask $\mathbf{m}$. It should be noted that the non-candidate parameters (e.g., attention blocks) always remain frozen and are not updated at all.

**Weighted exponential smoothing with dual momentum**

After each gradient update step ($i$) for $\mathcal{M}_S$, parameters of $\mathcal{M}_T$ are updated by EMA of the student model parameters. Typically, EMA is governed by

$$\boldsymbol{\theta}_{i+1}^T = \delta\boldsymbol{\theta}_i^T + (1-\delta)\boldsymbol{\theta}_{i+1}^S, \tag{2}$$

where $\delta$ is the smoothing parameter. Further, it has been shown in (Tarvainen & Valpola, 2017a; Oquab et al., 2023; Koh et al., 2022) that setting $\delta$ to a high value (e.g., 0.998) maintains a stable teacher model that can be considered as a strong reference for past tasks $\{0, \ldots, t-1\}$. But updating the teacher model with a single smoothing parameter in cases where parameters are masked creates dissonance and increases forgetting because all the parameters are updated with equal importance, disregarding those parameters which are selected by the gradient scoring function (where $[\mathbf{m}_{ij} = 1]$). To account for masking, we modify Eq. 2 as

$$\boldsymbol{\theta}_{i+1}^T = \boldsymbol{p}\boldsymbol{\theta}_i^T + \boldsymbol{q}\boldsymbol{\theta}_{i+1}^S, \tag{3}$$

where $\boldsymbol{p}$ and $\boldsymbol{q}$ denote the smoothing parameters for the teacher and student model, respectively, and can be computed as

$$\begin{aligned} \boldsymbol{p} &= (\gamma - \delta)\mathbf{m} + \delta, \\ \boldsymbol{q} &= (\delta - \gamma)\mathbf{m} + 1 - \delta \end{aligned} \tag{4}$$

where $\gamma < \delta$. This implies that the selected parameters of the teacher model ($[\mathbf{m}_{ij} = 1]$) move slightly faster towards the student model as compared to the frozen candidate parameters (where $[\mathbf{m}_{ij} = 0]$). As such, parameters where $[\mathbf{m}_{ij} = 0]$ will move at a slow rate of $\delta$, and unmasked parameters would be updated with $\gamma$. When $\gamma = \delta$, the weighted scheme becomes EMA with a single smoothing parameter. A detailed proof is given in Appendix 8.1.

**Phase 2: Unsupervised Test Time Learning (TTL)**

After supervised training is completed, both $\mathcal{M}_T$ and $\mathcal{M}_S$ are deployed for Test Time Learning (TTL). We consider teacher ($\mathcal{M}_T$) and student ($\mathcal{M}_S$) models as two experts on different data distributions, the $\mathcal{M}_S$ on the most recent and the $\mathcal{M}_T$ on previous sessions distributions.

We take inspiration from Out-of-Distribution (OOD) literature (Hendrycks & Gimpel, 2016), where a predictor assigns high scores to In-Distribution (ID) samples. Recent work (Hendrycks et al., 2019) shows that using the unnormalized maximum logit as an ID score is more robust than softmax probability, which can overconfidently

classify unknown samples (Yang et al., 2021). For CLIP, this logit corresponds to the cosine similarity between the image batch and text features. Following (Hendrycks et al., 2019), we use the maximum logit value of each expert as an ID score and select for each test sample the expert with the highest ID score, indicating that the sample is likely to be better represented by said expert. We then accept the pseudo-label of the selected expert. Formally, the pseudo label can be calculated as follows:

$$\hat{y} = \begin{cases} \hat{y_T} & \text{if } l_T \geqslant l_S \\ \hat{y_S} & \text{otherwise} \end{cases}, \tag{5}$$

where $\hat{y}$ is the accepted pseudo label and $l_T = \max(\mathcal{M}_T(\mathbf{x}))$ and $l_S = \max(\mathcal{M}_S(\mathbf{x}))$ are the maximum logit score for teacher and student model respectively, and similarly $\hat{y_T} = \arg\max(\mathcal{M}_T(\mathbf{x}))$ and $\hat{y_S} = \arg\max(\mathcal{M}_S(\mathbf{x}))$ are the pseudo labels by teacher and student models respectively. During test-time training, the student model $\mathcal{M}_S$ is updated by minimizing CLIP contrastive loss given a pseudo label $\hat{y}$. In realistic settings, multiple iterations on test data are often not always possible, for example, in a streaming data pipeline. We too mimic this setting, where the entire data is processed only once during the TTL phase.

Similar to the above-mentioned supervised phase, we also apply sparse local updates to $\mathcal{M}_S$. However, the estimation of masks based on the online data might be noisy and largely reduce the efficiency, as gradients of all parameters must be estimated for each mini-batch of test samples. To overcome this, and following the assumption that test data are drawn from the distributions of all previous tasks, we leverage the masks estimated for previous tasks[1]. We accumulate a union of the binary masks $(\mathbf{m}_u)$ over all the previously seen tasks $t$ such that $\mathbf{m}_u = \mathbf{m}_1 \cup \mathbf{m}_2 \cup \ldots \ldots \mathbf{m}_t$. To maintain the same sparsity level $(c = 0.1)$ of performed updates, we further select the same top-K (K=$c$) most relevant parameters from these new masked $\mathbf{m}_u$ parameters based on their previously computed gradient scores.

Finally, $\mathcal{M}_T(\boldsymbol{\theta}^T)$ is updated using the same dual momentum scheme, but with different smoothing vectors $\boldsymbol{p}', \boldsymbol{q}'$ as:

$$\boldsymbol{\theta}_{i+1}^T = \boldsymbol{p}' \boldsymbol{\theta}_i^T + \boldsymbol{q}' \boldsymbol{\theta}_{i+1}^S, \tag{6}$$

where $\boldsymbol{p}' = (\lambda - \delta)\mathbf{m} + \delta$ and $\boldsymbol{q}' = (\delta - \lambda)\mathbf{m} + 1 - \delta$. In the TTL phase, the momentum parameter $\lambda$ is kept such that $\gamma < \lambda < \delta$. Similar to phase 1, the selected parameters $[\mathbf{m}_u = 1]$ of Teacher Model $(\boldsymbol{\theta}^T)$ move slightly faster towards the student model $\boldsymbol{\theta}^S$ as compared to frozen candidate parameters $[\mathbf{m}_u = 0]$, ensuring stability in case of frequent and possibly noisy online updates. We discuss the effect of different momentum values in 8.2. The algorithm can be fully understood as given in algorithm block 1.

# 4 Experiments

## 4.1 Setup

**Architecture:** We apply DoSAPP to vision-language classification tasks, given their relatively robust knowledge measurement in such tasks. To ensure consistency across experiments, CLIP-ViT-B/16 (Radford et al., 2021) is used as the backbone in DoSAPP as well as in all the baselines. We report the accuracies recorded by the Teacher model. We refer to (Zhang et al., 2024) for hyperparameter selection other than dual momentum, which are given in Appendix 8.3.

**Datasets:** We consider five different vision datasets, three fine-grained (*Aircraft* (Maji et al., 2013), *CUB* (Wah et al., 2011), *Stanford Cars* (Krause et al., 2013), one coarse dataset (*CIFAR100* (Krizhevsky, 2012)) and one out-of-distribution dataset (*GTSRB* (Stallkamp et al., 2012)). These datasets are chosen primarily based on their initially low zero-shot performance with CLIP pre-trained models. To form the continual learning sequences, we split each dataset into 10 subsets with disjoint classes, composing 10 tasks. For all the datasets, the training data is used in the supervised learning phase. The test data is divided into two disjoint splits, $\mathcal{D}^u$ and $\mathcal{D}^e$, where $\mathcal{D}^u$ is used for unsupervised test-time learning and $\mathcal{D}^e$ is reserved for evaluation. This separation ensures a fair assessment of the method's generalization performance.

**Evaluation Metrics:** After each supervised session $t_i$ and the following test-time adaptation session, we evaluate the model's test performance on holdout datasets from all $T$ tasks. To do this, we construct the

---

[1]Parameters not relevant to the current stream of tasks will remain frozen, maintaining models' unrelated generic knowledge.

Table 2: Acc. (Average Accuracy, ↑) and F. (Forgetting, ↓) of different methods all using CLIP ViT-B/16 backbone with trainable vision and text encoders, without any Replay Buffer in CIL scenario. DoSAPP can achieve positive backward transfer: forgetting is negative on the Cars data (without ER).

| Method | Aircraft | | Cars | | CIFAR100 | | CUB | | GTSRB | |
|---|---|---|---|---|---|---|---|---|---|---|
| | Acc. (↑) | F. (↓) | Acc. (↑) | F. (↓) | Acc. (↑) | F. (↓) | Acc. (↑) | F. (↓) | Acc. (↑) | F. (↓) |
| CLIP-Zeroshot (Radford et al., 2021) | 24.45 | - | 64.63 | - | 68.25 | - | 55.13 | - | 43.38 | - |
| FLYP (fine-tuning) (Goyal et al., 2023) | 18.63 | 39.93 | 51.64 | 25.65 | 46.26 | 37.78 | 45.74 | 26.62 | 21.76 | 55.48 |
| MAS (Aljundi et al., 2018) | 33.69 | 27.50 | 69.43 | 9.18 | 63.88 | 21.16 | 61.72 | 12.05 | 42.04 | 25.38 |
| ZSCL (Zheng et al., 2023) | 30.96 | 15.65 | 67.79 | 8.27 | **80.50** | **1.05** | 61.09 | 7.69 | 62.92 | 13.54 |
| SPU (Zhang et al., 2024) | 30.94 | 28.36 | 69.41 | 16.91 | 58.80 | 26.37 | 62.31 | 7.2 | 43.06 | 19.16 |
| **DoSAPP** | **39.14** | **12.55** | **74.87** | **-0.74** | 79.16 | 7.73 | **68.17** | **2.15** | **72.33** | **1.02** |
| ER methods (ER=1000) | | | | | | | | | | |
| FLYP (fine-tuning) + ER (French, 1999) | 41.42 | 31.38 | 69.08 | 16.42 | 82.86 | 3.41 | 64.07 | 17.72 | **96.28** | **-7.48** |
| LWF + ER (Li & Hoiem, 2017) | 36.08 | 18.12 | 72.56 | 4.04 | 74.32 | 8.16 | 65.11 | 5.90 | 53.56 | 11.86 |
| PRD + ER (Asadi et al., 2023) | 37.11 | 17.35 | 74.08 | **3.75** | 79.66 | 3.10 | 65.92 | 6.55 | 63.00 | 12.44 |
| L2P + ER (Wang et al., 2022d) | 32.20 | 21.73 | 67.04 | 11.22 | 67.71 | 18.81 | 64.04 | 6.82 | 75.45 | 2.68 |
| DualPrompt + ER (Wang et al., 2022c) | 26.61 | 17.20 | 63.30 | 18.67 | 61.72 | 19.87 | 64.38 | 12.94 | 69.65 | 8.43 |
| SparseCL + ER (Wang et al., 2022b) | 31.95 | 19.77 | 71.57 | 5.38 | 69.35 | 15.23 | 62.50 | 9.66 | 48.99 | 24.91 |
| SLCA + ER (Zhang et al., 2023) | 29.40 | 11.45 | 62.65 | 4.42 | 70.03 | 0.19 | 53.87 | 7.75 | 46.01 | 0.83 |
| SPU + ER | 42.89 | 15.55 | 73.69 | 5.84 | 79.65 | 7.36 | 71.92 | 4.67 | 87.64 | 2.18 |
| **DoSAPP + ER=200** | **47.32** | **8.10** | **79.17** | 3.92 | **88.41** | **-1.96** | **74.39** | **2.77** | 83.67 | 1.92 |

matrix $R \in \mathbb{R}^{T \times T}$, where $R_{i,j}$ is the test classification accuracy of the model on task $t_j$ after observing the last sample from task $t_i$. Thus, we compute **Average Accuracy** (Acc. $= \frac{1}{T} \sum_{i=1}^{T} R_{T,i}$), **Average Forgetting** (F. $= \frac{1}{T-1} \sum_{j=1}^{T-1} f_{j,T}$, $f_{j,T} = \max_{i \in \{1,...,T-1\}}(a_{i,j} - a_{T,j})$) (Lopez-Paz & Ranzato, 2017; Wang et al., 2024a; Zhang et al., 2024). These metrics allow us to assess how well a continual learner solves a classification problem while overcoming forgetting.

**Baselines:** We comprehensively compare our method against various baselines. Firstly, we evaluate our approach against the best fine-tuning method of CLIP, FLYP (Goyal et al., 2023). We further integrate FLYP classical continual learning components to evaluate their performance on the CLIP backbone, including ER (Rolnick et al., 2019), weight regularization method, MAS (Aljundi et al., 2018), and functional regularization methods LwF (Li & Hoiem, 2017) and PRD (Asadi et al., 2023). We combine these functional regularization methods with a replay buffer (ER). We further consider the latest pre-trained model-based continual learning techniques L2P (Wang et al., 2022d), DualPrompt (Wang et al., 2022c), and SLCA (Zhang et al., 2023). Finally, we compare to recent methods that target knowledge retention of foundation models ZSCL (Zheng et al., 2023), SparseCL (Wang et al., 2022b), and SPU (Zhang et al., 2024).

# 5 Results

## 5.1 Comparison with CL Methods

We compare our method (DoSAPP) with recent and diverse approaches in the challenging scenario of class incremental learning (CIL), as shown in Table 2. DoSAPP achieves the largest improvement over simple fine-tuning (FLYP) (Goyal et al., 2023). Compared to MAS (Aljundi et al., 2018), DoSAPP increases performance by at least 16% on Cifar100, highlighting that not all parameters require updating. DoSAPP also outperforms prompt-based methods L2P (Wang et al., 2022d) and DualPrompt (Wang et al., 2022c) by 12% and 17% respectively on Cifar100. Even against dynamic network approaches like SLCA (Zhang et al., 2023), SparceCL (Wang et al., 2022b), ZSCL (Zheng et al., 2023), and SPU (Zhang et al., 2024), DoSAPP achieves state-of-the-art or comparable results across all datasets. This demonstrates that test-time data can enhance transferability and preserve learned knowledge. Notably, DoSAPP achieves strong performance without a replay buffer, and when provided a small buffer of just 20 samples per task, it significantly outperforms methods requiring large buffers.

Table 3: Acc. (Average Accuracy, ↑) and F. (Forgetting, ↓) in evaluating DoSAPP on imbalanced test data (referred as imb. in the table below), demonstrating its effectiveness in mitigating forgetting while maintaining high accuracy.

| Method | Aircraft | | Cars | | CIFAR100 | | CUB | | GTSRB | |
|---|---|---|---|---|---|---|---|---|---|---|
| | Acc. (↑) | F. (↓) | Acc. (↑) | F. (↓) | Acc. (↑) | F. (↓) | Acc. (↑) | F. (↓) | Acc. (↑) | F. (↓) |
| FLYP (finetune) | 16.20 | 42.85 | 47.92 | 29.40 | 43.30 | 40.12 | 41.63 | 29.61 | 18.52 | 58.70 |
| ZSCL | 26.89 | 23.80 | 64.58 | 10.85 | 62.10 | 12.67 | 57.77 | 10.54 | 58.45 | 17.91 |
| SLCA | 27.44 | 19.23 | 56.83 | 14.17 | 59.92 | 18.01 | 48.66 | 10.82 | 39.67 | 21.56 |
| SPU | 30.94 | 28.36 | 69.41 | 16.91 | 58.80 | 26.37 | 62.31 | 7.2 | 43.06 | 19.16 |
| **DoSAPP** | **35.99** | **15.26** | **72.68** | **6.38** | **75.70** | **9.81** | **64.84** | **3.73** | **68.17** | **5.63** |

Table 4: Acc. (Average Accuracy, ↑) and F. (Forgetting, ↓) comparing DoSAPP with CIL methods like SPU (Zhang et al., 2024), SparsCL (Wang et al., 2022b) integrated with one of the most recent CTTA method: RMT (Döbler et al., 2023). It can be observed that fusing the typical CTTA method in the CIL pipeline exacerbates the catastrophic forgetting. DoSAPP, on the other hand, outperforms all of them by a significant margin on all the datasets.

| Method | Aircraft | | Cars | | CIFAR100 | | CUB | | GTSRB | |
|---|---|---|---|---|---|---|---|---|---|---|
| | Acc. (↑) | F. (↓) | Acc. (↑) | F. (↓) | Acc. (↑) | F. (↓) | Acc. (↑) | F. (↓) | Acc. (↑) | F. (↓) |
| SPU | 30.94 | 28.36 | 69.41 | 16.91 | 58.80 | 26.37 | 62.31 | 7.2 | 43.06 | 19.16 |
| SPU + $D^u$ | 27.72 | 24.86 | 68.91 | 7.34 | 74.09 | 10.43 | 61.21 | 4.01 | 60.17 | 6.94 |
| SparsCL + RMT | 27.11 | 16.29 | 69.81 | 17.22 | 70.82 | 12.25 | 60.03 | 10.58 | 51.98 | 11.40 |
| SPU + RMT | 29.33 | 15.10 | 62.32 | 21.95 | 63.06 | 23.28 | 63.87 | 6.34 | 54.13 | 17.56 |
| **DoSAPP** | **39.14** | **12.55** | **74.87** | **-0.74** | **79.16** | **7.73** | **68.17** | **2.15** | **72.33** | **1.02** |

**Imbalanced test data:** We evaluate DoSAPP under a more realistic and challenging setting where $D_u$ (unsupervised data) is class-imbalanced as shown in Table 3. Each task is sampled from a symmetric Dirichlet distribution with a concentration parameter equal to the task length, leading to severe class imbalance, and at times, the complete absence of certain classes. This setup mirrors real-world scenarios where test-time data is often skewed. For fair evaluation, we retain a balanced evaluation set. Remarkably, despite the imbalance, DoSAPP still yields significant gains over purely supervised CL methods. While its performance is lower than in the balanced setting, this is expected: the model naturally adapts more to frequently observed classes during TTL. In practice, such behavior is desirable, as performance on the test distribution, rather than on rarely observed classes, is often the priority at deployment.

## 5.2 Comparison with TTA+CL Methods

We highlight the key innovation of our approach: *leveraging unsupervised test-time data - readily available in production, to enhance continual learning.* In contrast, most existing CL methods are not designed to incorporate such data, limiting their adaptability in this setting. Here, we investigate whether approaches that leverage unsupervised data in an online setting are capable of fully benefiting from test-time data in our setting. To examine this, we combine the best-performing CL method, SPU, with a simple self-training mechanism (SPU + $D^u$), where pseudo-labels are assigned based on the model's max logit output. We also integrate RMT (Döbler et al., 2023), with SPU (Zhang et al., 2024) and SparsCL (Wang et al., 2022b). Although more recent CTTA methods like DSS (Wang et al., 2024b) and PSMT (Tian & Lyu, 2024) have been proposed, we select RMT due to its consistently lower mean classification error across all the benchmark corruption datasets, making it a stronger baseline. We find that our proposed method, DoSAPP, consistently outperforms all variants as shown in Table 4. This highlights a critical limitation: *Continual Test Time Adaptation (CTTA) methods, when combined with ongoing supervised learning, suffer from substantial forgetting due to their inability to adapt across a long sequence of shifting tasks.* DoSAPP addresses this with a principled use of dual momentum over masked parameters. Furthermore, RMT performs worse than basic pseudo-labeling in nearly all cases, confirming that TTA methods are ill-suited for continual learning with expanding task distributions. We also demonstrate in Appendix 8.6 and 8.8 that DoSAPP remains robust to noise in test-time data and scales effectively with varying amounts of unlabeled input.

Table 5: Average Accuracy (Avg Acc.), First Task Accuracy (FTA), Current Task Accuracy (CTA), and Forgetting (F.) measured for a long sequence tasks made by the concatenation of the Aircraft (Maji et al., 2013) and Cars (Krause et al., 2013) datasets.

| Method (CLIP) | Avg Acc. ($\uparrow$) | FTA ($\uparrow$) | CTA ($\uparrow$) | F. ($\downarrow$) |
|---|---|---|---|---|
| Finetuning (no TTL) | 35.24 | 5.90 | **75.44** | 16.87 |
| SPU | 39.62 | 24.31 | 74.94 | 7.32 |
| **DoSAPP** | **45.01** | **30.63** | 71.13 | **2.34** |

Table 6: Acc. (Average Accuracy, $\uparrow$) and F. (Forgetting, $\downarrow$) when different components of DoSAPP are incrementally added to the Teacher-Student framework referred to as A1. A2 denotes the sparse parameter selection added to A1. EMA ($\delta$) represents single momentum updates, while EMA ($\delta, \gamma$) refers to dual momentum updates. $\mathbf{m}_u$ denotes the union of mask technique described in section 3.1.

| ID | Description | Aircraft | | Cars | | CIFAR100 | | CUB | | GTSRB | |
|---|---|---|---|---|---|---|---|---|---|---|---|
| | | Acc. ($\uparrow$) | F. ($\downarrow$) | Acc. ($\uparrow$) | F. ($\downarrow$) | Acc. ($\uparrow$) | F. ($\downarrow$) | Acc. ($\uparrow$) | F. ($\downarrow$) | Acc. ($\uparrow$) | F. ($\downarrow$) |
| A1 | Teacher-Student (EMA ($\delta$)) | 30.12 | 13.50 | 67.72 | 3.66 | 77.82 | **5.17** | 62.67 | 4.11 | 53.57 | 5.38 |
| A2 | A1 + sparse params | 34.16 | 18.61 | 69.42 | 3.41 | 71.93 | 8.24 | 66.32 | 3.98 | 55.32 | 5.81 |
| A3 | A2 + $\mathbf{m}_u$ | 31.79 | 10.42 | 70.99 | 3.64 | 72.66 | 8.86 | 66.98 | 3.17 | 61.54 | 4.01 |
| A4 | A2 + EMA ($\delta, \gamma$) | 35.49 | 11.53 | 72.14 | 3.58 | 75.93 | 8.02 | 67.28 | 3.41 | 64.15 | 3.20 |
| **DoSAPP*** | **A4 + $\mathbf{m}_u$** | **39.14** | **12.55** | **74.87** | **-0.74** | **79.16** | 7.73 | **68.17** | **2.15** | **72.33** | **1.02** |

## 5.3 Class Incremental Long Sequence scenario with domain shift

We consider a long sequence of tasks trained in a class-incremental manner. For these experiments, we combined 10 tasks from Aircraft (Maji et al., 2013) and 10 from Cars (Krause et al., 2013), introducing a domain shift after the first 10 tasks. Table 5 shows that our method, DoSAPP, outperforms SPU and Finetuning. Unlike SPU and Finetuning, which exhibit recency bias, DoSAPP achieves better overall and first-task accuracy, with only a 3.8% drop in current task accuracy (CTA). This demonstrates its ability to retain earlier knowledge while adapting effectively to new tasks. We also report per task forgetting for Finetuning, SPU, and DoSAPP highlighting the effectiveness of our proposed method in subsection 8.7.

## 6 Ablation Study

In this section, we quantitatively analyze the effect of different components of our proposed method, DoSAPP. We evaluate the effects of each component incrementally, as seen in Table 6. We begin with a baseline Teacher-Student setup using a single momentum (**A1**), and compare it with a variant that applies localized sparse updates to the first MLP layer of each transformer block (**A2**). This yields performance gains in 4 out of 5 datasets. Next, we introduce the union of supervised task masks ($\mathbf{m}_u$) during the TTL phase (**A3**), which leads to marginal improvements across most datasets, except for Aircraft—where applying a single momentum to both masked and unmasked parameters causes a mismatch and accuracy drop. To isolate the effect of dual momentum, we incorporate it into A2 without $\mathbf{m}_u$ (**A4**), which shows consistent improvements over both **A2** and **A3**. Finally, combining both dual momentum and mask union yields our complete method, **DoSAPP**, which achieves the best overall performance. These results underline the individual value of each component and their synergistic effect when integrated. Additional ablations on the sparsity threshold $c$, importance of pseudo label selection from both Teacher-Student, and the role of TTL phases are provided in Appendices 8.9, 8.10 and 8.11.

## 7 Discussion and Conclusion

This work explores how test-time data can be leveraged to improve the retention of previous tasks, drawing inspiration from human learning to build more adaptive models. We show that when used effectively, test-time data is a valuable resource. Our method, DoSAPP, enhances CLIP's zero-shot performance in the challenging class-incremental setting by combining sparse parameter updates with dual-momentum EMA across supervised

and unsupervised phases. Although our experiments focus exclusively on CIL classification, future work could investigate whether DoSAPP's mechanisms generalize to generative or other transformer-based tasks.

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

# 8 Appendix / supplemental material

## 8.1 Derivation for dual momentum

In section 3, the teacher model parameters $\boldsymbol{\theta}_i^T$ undergo exponential moving average as

$$\boldsymbol{\theta}_{i+1}^T = \boldsymbol{p}\boldsymbol{\theta}_i^T + \boldsymbol{q}\boldsymbol{\theta}_{i+1}^S \tag{7}$$

where $\boldsymbol{p}$ and $\boldsymbol{q}$ denote the smoothing parameters for the teacher and student model, respectively, and can be computed as

$$\begin{aligned} \boldsymbol{p} &= \alpha_1 \mathbf{m} + \beta_1, \\ \boldsymbol{p} &= \alpha_2 \mathbf{m} + \beta_2 \end{aligned} \tag{8}$$

where $\alpha_i$ and $\beta_i$ for $i \in \{1, 2\}$ are the coefficients for the affine transformation of the boolean mask vector $\mathbf{m}$.

To account for masked parameters, two momentum values $\delta, \gamma$ are introduced for teacher and student models respectively, such that for the teacher model, affine coefficients $\alpha_1$, $\beta_1$ are computed by solving the equations:

$$\alpha_1[\mathbf{m}_{ij} = 1] + \beta_1 = \gamma \ , \qquad \alpha_1[\mathbf{m}_{ij} = 0] + \beta_1 = \delta \tag{9}$$

and $\alpha_2, \beta_2$ are computed by solving the equations

$$\alpha_2[\mathbf{m}_{ij} = 1] + \beta_2 = 1 - \gamma \ , \qquad \alpha_2[\mathbf{m}_{ij} = 0] + \beta_2 = 1 - \delta \tag{10}$$

This gives

$$\begin{aligned} \alpha_1 &= \gamma - \delta, \quad \beta_1 = \delta \ , \\ \alpha_2 &= \delta - \gamma, \quad \beta_2 = 1 - \delta \end{aligned} \tag{11}$$

This gives

$$\begin{aligned} \boldsymbol{p} &= (\gamma - \delta)\mathbf{m} + \delta \ , \\ \boldsymbol{q} &= (\delta - \gamma)\mathbf{m} + 1 - \delta \end{aligned} \tag{12}$$

## 8.2 Effect of Momentum $(\gamma, \lambda)$ on Average Accuracy

Table 7 shows the sensitivity of our method on the choice of momentum values $\lambda, \delta$ in Table 7. A high $\delta$ has been chosen to keep the Teacher model stable, as shown in (Tarvainen & Valpola, 2017a; Oquab et al., 2023; Koh et al., 2022). It can be seen that when $\gamma = \lambda$ (single momentum EMA), the performance significantly drops. DoSAPP is less sensitive to the choice of $\gamma$, but it highly depends on $\lambda$. We can also see that as $\lambda < \gamma$, the performance again drops.

## 8.3 Hyperparameters

Table 8 shows different hyperparameters that have been used for all the experiments using CLIP backbones. The hyperparameters were selected based on the performance of the first task of the Cars (Krause et al., 2013) dataset. All the results have been gathered over experiments running on Nvidia V100 GPU, averaged over 5 random seeds.

## 8.4 Limitation

DoSAPP is a robust algorithm which can be potentially applied to any CL technique for unsupervised adaptation of Test Time Data. However, since it utilises the test data, its primary bottleneck becomes the quality of test data especially if its highly skewed. Another limitation is the increase in the computational

Table 7: Effect of Momentum $(\gamma, \lambda)$ on Average Accuracy (Acc in % ), Average Forgetting (F.), and First Task Accuracy (FTA.) *0.9999, 0.8, 0.9 have been used in the main results.

| Momentum $(\gamma, \lambda)$ | Aircraft | | |
|---|---|---|---|
| | Acc. ($\uparrow$) | F. ($\downarrow$) | FTA. ($\uparrow$) |
| 0.9999, 0.9999 | 23.99 | 18.36 | 12.15 |
| 0.5, 0.9 | 38.41 | 13.27 | 37.64 |
| 0.7, 0.9 | 37.22 | 13.05 | 37.72 |
| 0.8, 0.9* | **39.14** | **12.55** | **38.13** |
| 0.8, 0.6 | 37.06 | 15.12 | 29.63 |
| 0.8, 0.5 | 32.95 | 13.40 | 26.33 |

Table 8: Hyper Parameters for all the experiments using the CLIP ViT-B/16 model.

| Hparams | CLIP model |
|---|---|
| Batch Size | 64 |
| Optimizer | AdamW |
| Learning Rate | $7.5e-6$ |
| CL Epochs | 10 |
| Buffer | 0 |
| TTL batch size | 64 |
| Momentum-EMA $(\delta, \gamma, \lambda)$ | 0.9999, 0.8, 0.9 |
| sparsity ($c$) | 0.1 |

budget due to two deployed models: Teacher-Student framework. We address this by leveraging efficient sparse parameter selection method. While DoSAPP remains effective under class imbalance Table 3, extreme skew or partial coverage in the test-time stream may still bias updates toward overrepresented classes. To improve robustness in such settings, several practical mitigations can be incorporated without modifying the core framework. First, confidence-based update gating can prevent updates when both teacher and student produce low-confidence predictions, reducing drift caused by unreliable pseudo-labels. Second, a balanced mask reactivation strategy can periodically reintroduce a small portion of past-task mask entries based on earlier gradient scores, stabilizing parameters corresponding to tasks that are temporarily absent. Third, slower EMA updates (using a $\lambda$ closer to $\delta$ ) can reduce teacher drift when the observed stream is highly imbalanced. Finally, adaptive learning-rate scaling based on prediction entropy can temper the influence of majority-class samples during TTL. These mechanisms are lightweight, compatible with DoSAPP's sparse-update design, and improve applicability in real-world settings where test-time data is often skewed.

## 8.5 Broader Impact

This work proposes a general algorithm DoSAPP which can be implemented in any of the existing continual learning settings. Further it can be also integrated with other CL algorithms. Therefore, the potential negative societal impacts of our method are similar to those of the Continual Learning algorithms. Generally, DoSAPP would greatly improve performance especially in Test Time Learning. However, as with most CL algorithms, DoSAPP cannot guarantee to take safe and effective actions in all kinds of scenarios. We advocate that the users of DoSAPP should be aware of the potential consequences and utilize DoSAPP safely, especially in online environments such as self-driving, robotics, and healthcare.

## 8.6 Dependence on quality of test data used for unsupervised learning

We want to highlight that the trained model is expected to generalize to the distribution of the test data. We also assume that any quality degradation will be consistent across time steps. For instance, if the data is corrupted with noise, our method would generalize and adapt the model to this corruption as well. To

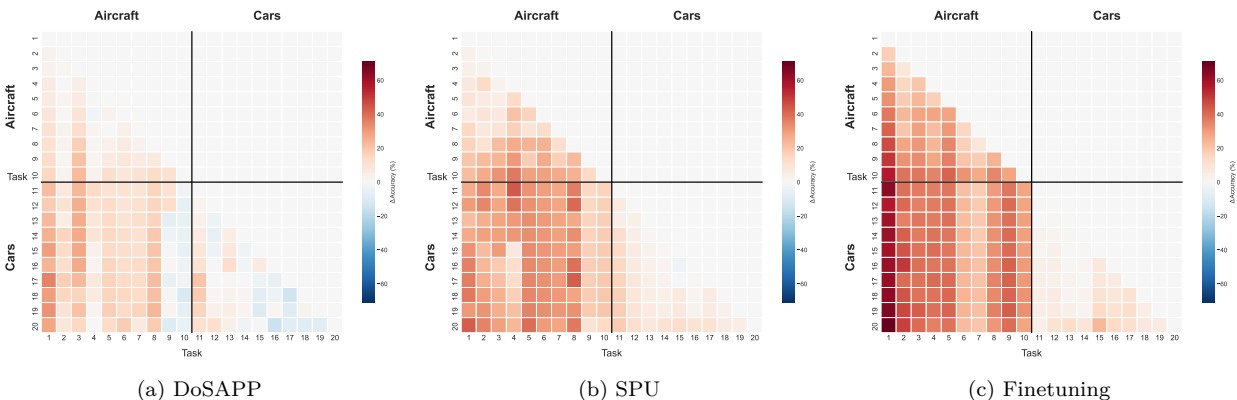

(a) DoSAPP  (b) SPU  (c) Finetuning

Figure 3: **Per-task forgetting matrices** for the long-sequence CIL setting (Aircraft → Cars). Each heatmap shows $F_{t,i} = R_{i,i} - R_{t,i}$, i.e., how much performance on task $i$ is lost after learning later tasks. The vertical and horizontal black lines denote the domain shift from Aircraft (left/top) to Cars (right/bottom). ***DoSAPP*** achieves the lowest forgetting across the entire task sequence, with several tasks even exhibiting *negative forgetting*, indicating improved retention as training progresses. In contrast, SPU shows moderate degradation, while standard finetuning undergoes severe catastrophic forgetting in both domains.

illustrate this, we conducted a small experiment by adding random Gaussian noise (mean = 0, std = 0.1) to different combinations of the test and evaluation suite (referred to as GN in Table 9). The results are shown below, with average accuracy (Acc.) followed by forgetting (F.). We observe that when corruption is present in the test-time data, the model is still able to leverage these data and improve on clean evaluation data compared to the test-time baseline by a significant margin of 17% (SPU alone). Interestingly, the model adapted to test-time data with Gaussian noise performs better on evaluation data with Gaussian noise than the case when the test-time data is clean. This is evidence of our method's ability to adapt and generalize to the present test-time conditions.

Table 9: Performance of DoSAPP with noise added to $D^u$ and $D^e$ for CIFAR100 Data

| Test Time Data ($D^u$) | Evaluation Data ($D^e$) | Acc. ($\uparrow$) | F. ($\downarrow$) |
|---|---|---|---|
| Clean | Clean | 79.16 | 7.73 |
| GN | Clean | 75.67 | 9.93 |
| Clean | GN | 69.50 | 12.86 |
| GN | GN | 73.42 | 6.86 |

### 8.7 Forgetting in Long Sequence scenario with domain shift

To analyze forgetting across the long sequence setting (Aircraft→Cars), we compute the *per-task forgetting matrix* $F \in \mathbb{R}^{T \times T}$ from the full accuracy trajectory $R$, where $R_{t,i}$ denotes the test accuracy on task $i$ after learning task $t$. Following standard continual learning analysis, we define the forgetting of task $i$ after learning task $t > i$ as

$$F_{t,i} = R_{i,i} - R_{t,i}, \tag{13}$$

where $R_{i,i}$ is the accuracy of task $i$ at the moment it is first learned. Thus, $F_{t,i}$ quantifies the absolute loss in performance on task $i$ caused by subsequent tasks. We visualize $F$ using heatmaps in Fig. 3, with a block boundary at the Aircraft–Cars transition.

Across all methods, the domain shift from Aircraft (tasks 1–10) to Cars (tasks 11–20) induces substantial forgetting, but the magnitude varies sharply by algorithm. **DoSAPP exhibits markedly lower forgetting** throughout both domains, with only mild degradation in the upper-right block corresponding to cross-domain interference. **SPU shows moderate forgetting**, particularly within the Aircraft block, but remains substantially more stable than naive fine-tuning. In contrast, **Finetuning catastrophically forgets** earlier tasks, especially after the domain shift, as reflected by large negative values across most of the forgetting

matrix. These visualizations reinforce the quantitative results in Table 5: DoSAPP most effectively mitigates long-sequence and cross-domain forgetting.

## 8.8 Ablation study about the size of test-time data $D^u$

In our method, we divided the evaluation data into two halves. One half is for unsupervised learning ($D^u$), and the other half is for evaluation ($D^e$). In the table below, we feed the fraction of $D^u$ for test time learning. 0.25 means that 25% of the original $D^u$ is fed to the model for unsupervised learning. We notice that when the fraction is below 0.75, there is an appreciable difference between the performance of our proposed model. However, at 0.75, the performance is quite close to that of the whole $D^u$.

Table 10: Dependence of the performance of DoSAPP with different proportions of the testing data $D^u$ on the CIFAR100 dataset.

| Fraction of $D^u$ | Acc. ($\uparrow$) | F. ($\downarrow$) |
|---|---|---|
| 0.25 | 73.97 | 14.23 |
| 0.5 | 76.83 | 9.44 |
| 0.75 | 79.02 | 8.16 |
| 1 | 79.16 | 7.73 |

## 8.9 Ablation on sparsity threshold ($c$)

We conduct an ablation study on the sparsity threshold ($c$) on different datasets, as shown in Table 11. We observe that increasing the sparsity threshold (c) beyond 0.1 leads to modest gains, indicating parameter redundancy as the number of updated parameters increases. Conversely, drastically reducing it to 0.01 results in a significant drop in accuracy. This is because the small number of updated parameters do not have enough capacity to accommodate the learned tasks.

Table 11: Effect of Sparsity Threshold ($c$) on Average Accuracy Across Datasets

| Sparsity ($c$) | Aircraft | Cars | CIFAR100 | CUB | GTSRB |
|---|---|---|---|---|---|
| 0.01 | 35.52 | 70.12 | 72.31 | 63.04 | 68.25 |
| 0.05 | 37.98 | 72.84 | 77.42 | 65.91 | 70.43 |
| **0.10** | 39.14 | **74.87** | 79.16 | **68.17** | **72.33** |
| 0.30 | 40.48 | 74.12 | **80.91** | 67.54 | 71.88 |
| 0.50 | **41.49** | 73.65 | 78.44 | 66.82 | 70.02 |
| 0.90 | 41.34 | 72.97 | 77.85 | 65.93 | 70.31 |

## 8.10 Pseudo label only from teacher ($\mathcal{M}_T$) expert

It must be emphasized that pseudo-labeling is not solely dependent on the student model. We further perform an experiment where pseudo labels are given only by the teacher, and it can be observed that the performance across all the datasets deteriorates as shown in the Table 12. This is because the teacher model is purposefully updated with very high EMA momentum and adapts slowly to the most recent task. Its logits remain biased toward earlier tasks, leading to outdated pseudo-labels during TTL for most recent tasks at hand. The student, in contrast, is plastic and calibrated for the current task, making its logits critical for accurate pseudo-label selection. Our max-logit expert selection therefore balances teacher stability with student plasticity.

## 8.11 Ablation on TTL phase:

Below, we provide the ablation showing average accuracy on previously seen tasks (excluding the current task to avoid bias from supervised training) before and after the unsupervised test-time learning (TTL) phase,

Table 12: Acc. (Average Accuracy, ↑) and F. (Forgetting, ↓) comparing DoSAPP with configuration when only Teacher Expert is used in providing pseudo-label at TTL phase.

| Description | Aircraft | | Cars | | CIFAR100 | | CUB | | GTSRB | |
|---|---|---|---|---|---|---|---|---|---|---|
| | Acc. (↑) | F. (↓) | Acc. (↑) | F. (↓) | Acc. (↑) | F. (↓) | Acc. (↑) | F. (↓) | Acc. (↑) | F. (↓) |
| Pseudo-Label$_{\mathcal{M}_T}$ | 36.92 | 13.84 | 71.04 | 0.15 | 75.48 | 8.22 | 65.21 | 2.97 | 68.09 | 1.56 |
| **DoSAPP** | **39.14** | **12.55** | **74.87** | **-0.74** | **79.16** | **7.73** | **68.17** | **2.15** | **72.33** | **1.02** |

using the Aircraft dataset. The results show consistent improvement on previous tasks, demonstrating the usefulness of unsupervised TTL as observed in Table 13

Table 13: Average Test Accuracy (referred as Acc. below) Before and After TTL

| Task | Acc. (Before TTL) | Acc. (After TTL) |
|---|---|---|
| 1 | 78.03 | 78.03 |
| 2 | 66.55 | 68.21 |
| 3 | 57.32 | 60.13 |
| 4 | 50.67 | 54.09 |
| 5 | 45.13 | 46.22 |
| 6 | 43.39 | 47.58 |
| 7 | 39.21 | 45.47 |
| 8 | 41.84 | 45.08 |
| 9 | 41.97 | 44.14 |
| 10 | 37.38 | 39.80 |

We further provide an ablation in Table 14 showing average accuracy after training is complete on all the tasks using DoSAPP, with and without TTL phase. We can see that without the TTL phase, the model significantly deteriorates in performance, highlighting the usefulness of test time data.

Table 14: DoSAPP Ablation Study: With vs Without TTL

| Dataset | DoSAPP without TTL | DoSAPP with TTL |
|---|---|---|
| Aircraft | 35.42 | 39.14 |
| CIFAR100 | 67.30 | 79.16 |

## 8.12 Comparing Evaluation on full test set

Here we compared one of the SOTA CL baselines (SPU (Zhang et al., 2024)) evaluated on the complete test set, instead of the subset $D_e$. From table Table 15, we can observe that DoSAPP when evaluated on the $D_e$, outperforms model trained via SPU method and evaluated on full test set ($D_u \cup D_e$).

Table 15: Acc. (Average Accuracy, ↑) and F. (Forgetting, ↓) where the latest CL method SPU is evaluated on the complete test set, i.e, without splitting the test set into $D_u$ and $D_e$. DoSAPP is still evaluated on $D_e$ since it utilizes $D_u$ for unsupervised training, and including this subset for evaluation will be unfair. These results show that although the baseline performance of SPU has increased slightly as compared to the performance mentioned in section 5, DoSAPP outperforms both with and without buffer.

| Method | Aircraft | | Cars | | CIFAR100 | | CUB | | GTSRB | |
|---|---|---|---|---|---|---|---|---|---|---|
| | Acc. (↑) | F. (↓) | Acc. (↑) | F. (↓) | Acc. (↑) | F. (↓) | Acc. (↑) | F. (↓) | Acc. (↑) | F. (↓) |
| SPU | 32.51 | 24.74 | 70.59 | 14.26 | 64.98 | 19.74 | 62.43 | 6.89 | 48.97 | 15.51 |
| **DoSAPP** | **39.14** | **12.55** | **74.87** | **-0.74** | **79.16** | **7.73** | **68.17** | **2.15** | **72.33** | **1.02** |
| SPU + ER=1000 | 44.43 | 14.42 | 77.51 | **3.26** | 83.99 | -0.39 | 71.51 | 4.84 | **94.25** | **-7.87** |
| **DoSAPP + ER=200** | **47.32** | **8.10** | **79.17** | 3.92 | **88.41** | **-1.96** | **74.39** | 2.77 | 83.67 | 1.92 |

### 8.13 Computation time:

We include a comparison of wall-clock training time (in seconds) between the full model (all parameters trainable) and the sparse variant (c = 0.1) across 10 tasks and one of the latest SOTA Continual Test Time Adaptation method (RMT) that we have utilized in Table 5.2 to compare with our proposed algorithm. As shown in Table 16, the sparse variant yields consistent but modest reductions in training time per task. While these improvements are incremental rather than substantial, they demonstrate that restricting updates to a small set of parameters does not introduce additional computational overhead.

Table 16: Training Time Comparison Across Methods

| Task | All Params | DoSAPP ($c = 0.1$) | SPU+RMT |
|------|-----------|--------------------|---------|
| 1 | 46.61 | 45.69 | 47.13 |
| 2 | 70.83 | 68.96 | 71.27 |
| 3 | 77.75 | 75.90 | 78.14 |
| 4 | 89.55 | 86.92 | 90.07 |
| 5 | 101.60 | 98.81 | 102.18 |
| 6 | 112.19 | 108.93 | 112.86 |
| 7 | 118.28 | 114.71 | 119.03 |
| 8 | 130.49 | 127.40 | 131.21 |
| 9 | 142.59 | 139.38 | 143.22 |
| 10 | 154.01 | 149.65 | 154.77 |

