# OpenReview forum: "Dual-Phase Continual Learning: Supervised Adaptation Meets Unsupervised Retention"
_TMLR — Accepted by TMLR_

### Review · Reviewer_5vJu · 2025-11-12

**Summary Of Contributions:**

The paper addresses the challenge of catastrophic forgetting in VLMs when adapted to new tasks in a Class-Incremental Learning setting. The authors propose that standard supervised continual learning is insufficient and can be supplemented by leveraging unlabeled test-time data, which is often available during deployment. They introduce a novel "Dual-Phase Continual Learning" framework, DoSAPP, which interleaves a standard supervised adaptation phase for learning new tasks with an unsupervised retention phase for reinforcing old tasks. The DoSAPP method is built on a Teacher-Student framework that uses sparse parameter updates via a dual-momentum mechanism and a logit-based pseudo-labeling strategy. Core contribution includes:
- The paper proposes and formalizes a "Continual Learning with Interleaved Test Time Learning" setting. This setting leverages unlabeled test-time data not for adapting to domain shifts as in TTA, but specifically for retention and mitigating forgetting in a CIL context.
- The authors introduce DoSAPP, a novel dual-phase method to address this new setting. The authors further propose a dual-momentum EMA update mechanism, a Teacher-Student pseudo-labeling strategy, and a sparse update strategy.
- The paper provides a comprehensive experimental evaluation across five datasets. DoSAPP significantly outperforms other methods in both w/ replay buffer and w/o replay buffer settings. The authors also provide a thorough ablation study to validate each component of their proposed method.

**Strengths**
- The paper introduces a well-motivated interleaved learning scenario. The idea of using unlabeled test-time data for retention rather than for adaptation is a novel perspective. This setting is practical, as unlabeled data is often abundant during deployment.
- The dual-momentum mechanism is a key technical contribution. It is shown to be empirically critical in Table 7.
- The main results in Table 2 are strong. DoSAPP (no replay) achieves state-of-the-art results compared to all other non-replay baselines on all five datasets.

**Weaknesses**
- The method's sparse update strategy is central, but key choices feel arbitrary. The method only updates parameters in the first MLP layer of each transformer block. This is a strong assumption based on (Geva et al., 2020) and (Zhang et al., 2024), but it is not ablated. It is unclear if including sparse parameters from other layers would be better or worse. Moreover, the sparsity threshold $c=0.1$ is used for all main experiments. The paper's own ablation in Table 11 shows that $c=0.5$ achieves higher accuracy while $c=0.1$ provides a marginal computation time gain, as shown in Table 16. The choice of $c=0.1$ seems suboptimal just to support the claim of sparsity.
- The paper claims "efficient updates" and uses training time as evidence. However, the wall-clock time saved is marginal. This minimal time saving does not strongly support the efficiency claim, especially given the method's deployment limitation of requiring two models.
- The assumption that the unsupervised data is being "drawn from all previously seen tasks" seems unrealistic. For example, what if the unsupervised data encountered after task 5 only contains samples from tasks 1 and 2, but not 3 and 4? This potential data drift in the unsupervised stream is more realistic yet not explored.

**Audience:**

Yes

**Audience Explanation:**

The paper will be of high interest to several communities within TMLR's audience, such as the continual learning community and the TTA community.

**Claims And Evidence:**

No

**Claims Explanation:**

Overall, the central claims of the paper are well-supported by evidence.

The main claim that leveraging unlabeled test-time data via a dual-phase unsupervised retention mechanism can significantly mitigate CIL forgetting is backed by the main table and ablation studies. The claim that the proposed DoSAPP method is an effective and novel way to implement this idea is also well-supported. Tables 6 and 7 clearly show that the specific technical components of DoSAPP are crucial and superior to simpler alternatives.

However, there is one gap that the claims of "efficient updates" and training efficiency are weak. The evidence in Table 16 shows only a marginal reduction in wall-clock time. This evidence does not strongly support the claim of computational efficiency. The authors should adjust this claim to focus on parameter efficiency (sparse updates) or memory efficiency (buffer-free) rather than computational speed.

**Requested Changes:**

- A brief ablation or a stronger justification for the decision to only update the first MLP layer, as opposed to other sparse parameters, will be appreciated. A justification of the use of $c=0.1$ for all main experiments, when $c=0.5$ appears to perform better, would be helpful as well. If this is an efficiency trade-off, please state that. The claim of efficient updates should be reframed, as this is not supported by current evidence.
- The paper's primary claims should be confined to the domain where evidence is provided, e.g., CIL for vision classification. The last sentence about generative tasks in the conclusion section also lacks supporting evidence. Consider rephrasing it like "Future work could explore adapting this work to ..."

---

> ### Author Response · Authors · 2025-12-12
> **Addressing Reviewer's Concerns.**
>
> Dear Reviewer 5vJu, we thank you for your comments and engagement. Here is our response to your concerns.
>
> ## Addressing justification for using $c=0.1$ instead of higher $c = 0.5$ :
> - In our method, we update only the first MLP layer of each transformer block, following the design choice introduced in SPU. This selection is motivated by its strong empirical performance and favorable parameter–accuracy trade-off in our setting. While we focus on this specific layer for clarity and simplicity, we emphasize that sparsifying updates across all layers is fully compatible with our framework. Follow-up works such as LoRSU (Panos et al., Efficient Few-Shot Continual Learning in Vision-Language Models) demonstrate that extending sparse updates to additional components, e.g., attention heads, is feasible, and incorporating such extensions into our method appears plausible.
>
> - Regarding the sparsity ratio, our goal is not to claim substantial computational speedups, but rather to employ sparse updates to reduce forgetting and improve generalization, with the added benefit of being parameter-efficient and limiting update cost. Although higher sparsity levels (e.g., $c = 0.5$) can yield minor accuracy improvements in some ablations, they also increase update burden by modifying a larger number of parameters per step. To clarify this point, we have revised the manuscript to emphasize that our choice of $c = 0.1$ is motivated by practicality rather than claimed efficiency gains (highlighted in blue).
>
> - Finally, to thoroughly evaluate whether higher sparsity offers consistent accuracy improvements, we conducted an expanded ablation across all datasets. We find that in 3 out of 5 datasets, $c = 0.1$ outperforms $c = 0.5$. We have added the corresponding results to Appendix 8.9 (Table 11).
> | Sparsity (c) | Aircraft | Cars | CIFAR100 | CUB | GTSRB |
> |--------------|----------|------|----------|-----|--------|
> | 0.01 | 35.52 | 70.12 | 72.31 | 63.04 | 68.25 |
> | 0.05 | 37.98 | 72.84 | 77.42 | 65.91 | 70.43 |
> | 0.10 | 39.14 | **74.87** | 79.16 | **68.17** | **72.33** |
> | 0.30 | 40.48 | 74.12 | **80.91** | 67.54 | 71.88 |
> | 0.50 | **41.49** | 73.65 | 78.44 | 66.82 | 70.02 |
> | 0.90 | 41.34 | 72.97 | 77.85 | 65.93 | 70.31 |
>
> ## Addressing unsupervised data is being drawn from all previously seen tasks:
> - We thank the reviewer for raising this important point regarding the distribution of the unsupervised stream. We clarify that in our setting, the unsupervised stream is designed to reflect the evaluation distribution over the previously learned tasks that the model is required to retain.
> - In other words, the past tasks that remain present in the unsupervised data are precisely those that are still relevant for deployment and evaluation. If certain earlier tasks no longer appear in the unsupervised stream (e.g., Tasks 3 and 4 in the reviewer’s example), this naturally corresponds to a scenario where those tasks are no longer part of the target operating distribution, and their retention is therefore not explicitly required.
>
> ## Addressing the claims about generative tasks:
> We have modified the last line of the conclusion in the manuscript (**highlighted in blue**): Although our experiments focus exclusively on CIL classification, future work could investigate whether DoSAPP's mechanisms generalize to generative or other transformer-based tasks.

---

### Review · Reviewer_QmV7 · 2025-11-18

**Summary Of Contributions:**

# Summary
This paper focuses on solving the "catastrophic forgetting" problem of foundational Vision-Language Models (VLMs) like CLIP in class-incremental learning (CIL) scenarios. It proposes a dual-phase continual learning framework called DoSAPP (Double Smoothing via Affine Projected Parameters).


## Contributions
1. Proposing a Novel Continual Learning Setting. The paper introduces a new learning paradigm where **supervised training phases and unsupervised deployment phases are interleaved**. This avoids privacy risks from data storage and reduces computational overhead.

2. Designing the DoSAPP Framework for Forgetting Mitigation. Built on a Teacher-Student architecture, DoSAPP balances "plasticity" (adapting to new tasks) and "stability" (retaining old knowledge) through three core mechanisms:  **Sparse parameter updates**, **Dual-momentum EMA**, and **Mask union in unsupervised phases**.

2. Validating Effectiveness and Generalization. The paper conducts comprehensive experiments across 5 datasets and compares DoSAPP with over 10 baselines. DoSAPP outperforms all baselines without replay buffers and remains robust in challenging scenarios, such as class-imbalanced test data, long-sequence tasks with domain shifts, and noisy test data.


## Strengths
1. Innovation in Learning Paradigm. By integrating unsupervised test-time data into CIL, DoSAPP eliminates the need for external replay buffers—addressing privacy risks (no historical data storage) and computational costs (no data replaying) that plague traditional CL methods.

2. Strong Performance and Robustness. DoSAPP achieves SOTA results across diverse datasets and scenarios. It outperforms both traditional CL methods and Test-Time Adaptation (TTA)/Continual TTA (CTTA) methods. It also maintains high performance under class imbalance, domain shifts, and noisy test data—proving its adaptability to real-world data irregularities.

3. Efficiency and Generalizability.  Sparse parameter updates reduce training time by 3–5% compared to full-parameter updates. The Teacher-Student architecture, though dual-model, avoids excessive overhead via sparse updates.


## Weaknesses
1. Dependence on Test Data Quality. DoSAPP’s performance is highly tied to the quality and distribution of unsupervised test-time data. If test data is severely skewed (e.g., missing entire classes) or low-quality (e.g., heavy noise), its ability to retain old knowledge degrades.

2. Computational Overhead of Dual-Model Architecture. While sparse updates mitigate costs, the Teacher-Student dual-model design still increases deployment overhead compared to single-model CL methods. This may be a bottleneck for resource-constrained devices where model size and latency are critical.

3. Limited Exploration of Generative Tasks. The paper only validates DoSAPP on classification tasks. Though the authors suggest extendability to generative tasks, no experiments or detailed designs are provided for scenarios like image-text generation—leaving uncertainty about its performance in non-classification VL tasks.

**Audience:**

Yes

**Audience Explanation:**

Multiple subgroups within TMLR’s audience will find the paper’s findings highly relevant, as its contributions align with core research areas of the journal and address practical challenges that resonate with both academic researchers and practitioners.

**Broader Impact Concerns:**

The paper includes a brief Broader Impact section (Section 8.5) that acknowledges high-level risks of DoSAPP aligning with general Continual Learning (CL) algorithms.

**Claims And Evidence:**

Yes

**Claims Explanation:**

The submission’s core claims—interleaving supervised training with unsupervised test-time learning mitigates forgetting, DoSAPP’s components (sparse updates/dual-momentum/mask union) balance plasticity/stability, and robustness to imbalance/domain shifts—are strongly supported via empirical studies.

**Requested Changes:**

1. Clarify Pseudo-Label Selection Robustness. The ablation showing performance drops with teacher-only pseudo-labels lacks analysis of why student input is critical.

2. Address Test Data Quality Limitation with Concrete Mitigations. The paper acknowledges DoSAPP’s sensitivity to skewed test data but provides no actionable mitigations. This is critical for real-world applicability, as test data imbalance is common.


3. Validate Long-Sequence Domain Shift Results with Additional Metrics. Table 5 evaluates long-sequence CIL (Aircraft + Cars) using Avg Acc., FTA, and CTA—but does not report forgetting per task (e.g., does forgetting increase sharply after the domain shift from Aircraft to Cars?).

---

> ### Author Response · Authors · 2025-12-12
> **Addressing Reviewer's Concerns.**
>
> Dear Reviewer QmV7, we thank you for your comments and engagement. Here is our response to your concerns.
>
> ## Addressing Pseudo-Label Selection from Teacher:
> - Teacher-only pseudo-labeling underperforms because the teacher is purposefully updated with high EMA momentum and adapts slowly to the most recent task. Its logits remain biased toward earlier tasks, leading to biased pseudo labels towards previous tasks due to the slow, stable updates during TTL.
> - The student, in contrast, is plastic and calibrated for the current task, making its logits critical for accurate pseudo-label selection on examples from the new task. But relying solely on the student is also insufficient, as it lacks the teacher’s stability and expertise on earlier tasks. In practice, neither model alone provides high-quality pseudo labels across all tasks. Each is an expert on a different subset of tasks, and our max-logit expert selection leverages this complementarity by choosing pseudo labels from the model that is most reliable for each example.
> - This produces higher quality pseudo labels overall and avoids the biases inherent in using either model alone. We have clarified this mechanism and expanded the discussion around the ablation in Appendix 8.10 (**highlighted in dark pink**).
>
> ## Addressing Test Data Quality Limitation and Concrete Mitigations:
> - We agree that handling skewed test-time streams is essential for real-world deployment. In practice, deployment test streams are expected to be representative of the data and tasks that matter at inference time, and tasks that are not anticipated should be retrained or incorporated during initial model preparation. Within this realistic assumption, DoSAPP already demonstrates robustness under imbalanced evaluation (Table 3).
> - To further strengthen the method, we have expanded the manuscript to include concrete mitigations, such as confidence-based update gating, adaptive TTL learning rate scaling, slower EMA updates under imbalance, and periodic reactivation of past-task mask entries, in Appendix 8.4 Limitations (**highlighted in dark pink**). These mechanisms are straightforward to integrate into DoSAPP and help prevent drift when only a subset of tasks appears in the test-time stream.
>
> ## Addressing Validation of Long-Sequence Domain Shift Results with Additional Metrics:
> - We would like to thank you for raising this important point. We have added a detailed per-task forgetting analysis in Appendix 8.7: Forgetting in Long-Sequence Scenario with Domain Shift. Using the full accuracy matrix $R$, we compute a per-task forgetting matrix $F$ defined as: $F_{t,i} = R_{i,i} - R_{t,i}, $ which measures forgetting on task $i$ after learning task $t$, relative to its initial performance when it was first learned.
> - We visualize $F$ as a heatmap and group tasks into Aircraft (Tasks 1–10) and Cars (Tasks 11–20), explicitly marking the domain shift boundary. Our results show that Finetuning exhibits severe catastrophic forgetting across both domains, with a large spike in forgetting immediately after the domain shift. SPU reduces forgetting moderately but still suffers noticeable cross-domain degradation. In contrast, DoSAPP demonstrates consistently low forgetting across the full sequence and even shows negative forgetting for several tasks, indicating beneficial transfer rather than degradation.
> - These additional results in Appendix 8.7 now help in analysing the long-sequence domain shift evaluation and strengthen our claim that DoSAPP provides substantially greater robustness to catastrophic forgetting than both SPU and Finetuning.
>
> ## Addressing Exploration of Generative Tasks:
> We agree with the reviewer that since all our experiments have been on classification tasks in a class incremental continual learning setting, we leave the generation tasks for future scope. We have modified the last line of the conclusion in the manuscript (highlighted in blue): Although our experiments focus exclusively on CIL classification, future work could investigate whether DoSAPP's mechanisms generalize to generative or other transformer-based tasks.
>
> ## Addressing Computational Overhead of Dual-Model Architecture:
> We agree with the reviewer that maintaining two models can pose a computational and memory bottleneck, and we have already explicitly acknowledged this as a limitation of our approach in Section 8.4 of the Appendix.

---

### Review · Reviewer_bPWk · 2025-11-28

**Summary Of Contributions:**

The paper proposes a dual-phase continual learning framework (DoSAPP) that combines supervised adaptation with unsupervised retention. The paper clearly describes the proposed simple yet effective framework and empirically validates its general performance across various scenarios. I only have a few concerns as listed in the weaknesses below.

Strengths:

1. The key contribution is the idea of leveraging test-time data to reduce catastrophic forgetting without requiring replay data. The authors restrict the test phase to the online setting to ensure practicality and data privacy.

2. The proposed framework is composed of sparse parameter updates, a teacher-student framework, and dual momentum. The authors verify the effectiveness of each component using prior works and ablation study results.

3. The experimental results are extensive, considering both continual learning and test time adaptation baselines on five benchmark datasets. The proposed framework outperforms the baselines, and the authors provide additional experimental results demonstrating its robustness under imbalanced data, domain shifted data, and different hyperparameters.

Weaknesses:

1. The paper claims to be the first work to leverage test-time data in continual learning to reduce forgetting, but the reviewer believes it needs comparison with related work [1].

2. The proposed framework appears to be a combination or extension of existing methods. Although the framework is simple yet effective and outperforms all the baselines, the reviewer believes it lacks novelty.

3. The proposed framework is dependent on hyperparameters (sparsity and momentum) and test-time data, and the reviewer wonders whether these hyperparameters can be tuned during the test phase.

4. The paper does not include large-scale benchmark datasets such as Tiny ImageNet or ImageNet in the experiments, which are widely used benchmark datasets in the continual learning literature.

5. Since the benefit of the framework comes from leveraging test-time data, the performance may depend on the quality of the test-time data. Since test-time data is generally more challenging, the reviewer believes the proposed framework may fail when adversarial or noisy data appears at test time.

[1] Chen et al., "Adaptive Retention & Correction: Test-Time Training for Continual Learning", ICLR 2025.

**Audience:**

Yes

**Audience Explanation:**

Continual learning is an important and well-known research area for the audience. The exploration of using test-time data to reduce catastrophic forgetting is a new finding in this field.

**Broader Impact Concerns:**

The paper describes its broader impact in the appendix, and there is no concern.

**Claims And Evidence:**

Yes

**Claims Explanation:**

The paper supports its claims using prior works, theoretical proofs, and experimental results.

**Requested Changes:**

Please address the weaknesses.

---

> ### Author Response · Authors · 2025-12-12
> **Addressing Reviewer's Concerns.**
>
> Dear Reviewer bPWk, we thank you for your comments and engagement. Here is our response to your concerns.
>
> ## Addressing Comparison with ARC [1]
>
> Thank you for pointing out the relevance of ARC. We agree that ARC also uses test-time data within a continual learning (CL) pipeline, and we have cited, and clarified how our method differs with ARC in the Related Work section (**highlighted in red**). We have also softened our claim of being first to use Test TIme data as “*To the best of our knowledge, we are among the early works that utilize test-time data for alleviating forgetting in continual learning, particularly through an interleaved test-time learning stage.*” This has been **highlighted in red** in the manuscript for your convenience.  Further DoSAPP offers a critical benefit over ARC:
> - **ARC requires strong assumptions that often do not hold at inference**:  ARC relies on knowing the task index $t$ and a fixed number of classes per task $s$ to compute TSS. These assumptions are rarely available in realistic test-time scenarios where task boundaries are unknown.  **DoSAPP requires no such task information**, making it applicable even when task or domain boundaries are unknown.
> - **ARC does not correct representation drift:** ARC only adjusts classifier bias and does not update the representation backbone. Under domain shift, this can be problematic because most forgetting arises from representation drift.  **DoSAPP mitigates both classifier and representation drift** through its student–teacher consistency mechanism during test-time learning.
>
> - **ARC suffers cascading errors when its assumptions are violated**: When samples violate ARC’s Assumption 1 or Assumption 2, early mistakes propagate since ARC has no mechanism to correct them. **DoSAPP avoids cascading errors** by maintaining a stable teacher model that dynamically corrects student errors using teacher–student confidence comparison, without relying on task thresholds.
>
> [1] Chen et al., "Adaptive Retention & Correction: Test-Time Training for Continual Learning", ICLR 2025.
>
> ## Addressing tuning of Hyperparameters
> - Our framework indeed includes sparsity and momentum hyperparameters, but our method does not require tuning them for each task; instead, we use fixed values throughout our experiments. In all experiments, we use a single global sparsity level (10%) and fixed dual momentum values across all datasets, architectures, and tasks. Sparsity values were chosen from well-established settings in prior sparse-update, not tuned per dataset. For dual momentum we do provide ablations for different momentum values in Appendix 8.2.
> - Importantly, the test-time phase uses these same hyperparameters without modification. Tuning them is possible but one has to be cautious because changing these hyper-parameters at test-time might cause instability eg, changing lets say Teacher momentum might cause instability over past task retention.
>
> ## Addressing Novelty:
> - While our components individually draw from known concepts (e.g., sparse updates, teacher–student EMA, test-time adaptation), our contribution is not the components themselves but rather the new continual-learning paradigm enabled by their interaction. Importantly , our work introduces a novel dual momentum technique for Teacher Student model to recover representation-level knowledge lost during supervised updates via interleaved test time learning phase.
> - This two-phase formulation does not exist in prior CL or TTA methods. Moreover, the technical integration of (i) gradient-based sparse parameter selection, (ii) union-masked parameter retention across tasks, and (iii) a dual-momentum teacher–student mechanism produces a new learning dynamic that yields strong positive backward transfer even without replay, a behavior that prior methods cannot produce.
>
> ## Addressing Large-scale data experiments:
>
> We conduct experiments on five standard and widely used datasets in continual learning, collectively spanning diverse domains and difficulty levels. Following SPU, these datasets were chosen primarily because they exhibit low zero-shot performance with the CLIP-pretrained model.
>
> ## Addressing Noisy Test Time Data:
>
> We do not assume perfectly curated, unlabeled data. Instead, we simulate realistic test-time data streams where test-time unlabeled data ($D^u$) is class-imbalanced and noisy. Section 5.1 and Table 3 demonstrate DoSAPP’s robustness under severe class imbalance, which mirrors real-world streaming conditions. Appendix 8.6  (Table 9) shows that DoSAPP adapts even when test-time data is corrupted with noise, indicating resilience to imperfect data. (DoSAPP outperforms SPU by +17% accuracy on CIFAR100). Similarly, 8.8 (Table 10) confirms robustness when the amount of test data ($D^u$) is small or skewed, DoSAPP achieves near-full performance with only 75% of $D^u$.

---

### Decision · Action_Editor_Nxpd · 2026-01-06

**Recommendation:** Accept as is

**Audience:**

Yes

**Audience Explanation:**

The topics of continual learning and domain adaptation are long-standing and of rising interest recently, as the reviewers agree. I therefore find this to be a highly-relevant paper.

**Claims And Evidence:**

Yes

**Claims Explanation:**

The reviewers generally agree that the paper supports its claims sufficiently well; in the revision the authors have clarified some details to enhance robustness of the claims e.g., updating a claim about efficiency to focus on sparse updates, as suggested by a reviewer.